# LF-Net: Learning Local Features from Images

**Yuki Ono**
Sony Imaging Products & Solutions Inc.
`yuki.ono@sony.com`

**Eduard Trulls**
École Polytechnique Fédérale de Lausanne
`eduard.trulls@epfl.ch`

**Pascal Fua**
École Polytechnique Fédérale de Lausanne
`pascal.fua@epfl.ch`

**Kwang Moo Yi**
Visual Computing Group, University of Victoria
`kyi@uvic.ca`

## Abstract

We present a novel deep architecture and a training strategy to learn a local feature pipeline from scratch, using collections of images without the need for human supervision. To do so we exploit depth and relative camera pose cues to create a virtual target that the network should achieve on one image, provided the outputs of the network for the other image. While this process is inherently non-differentiable, we show that we can optimize the network in a two-branch setup by confining it to one branch, while preserving differentiability in the other. We train our method on both indoor and outdoor datasets, with depth data from 3D sensors for the former, and depth estimates from an off-the-shelf Structure-from-Motion solution for the latter. Our models outperform the state of the art on sparse feature matching on both datasets, while running at 60+ fps for QVGA images.

## 1 Introduction

Establishing correspondences across images is at the heart of many Computer Vision algorithms, such as those for wide-baseline stereo, object detection, and image retrieval. With the emergence of SIFT [23], sparse methods that find interest points and then match them across images became the *de facto* standard. In recent years, many of these approaches have been revisited using deep nets [11, 33, 48, 49], which has also sparked a revival for dense matching [9, 43, 45, 52, 53].

However, dense methods tend to fail in complex scenes with occlusions [49], while sparse methods still suffer from severe limitations. Some can only train individual parts of the feature extraction pipeline [33] while others can be trained end-to-end but still require the output of hand-crafted detectors to initialize the training process [11, 48, 49]. For the former, reported gains in performance may fade away when they are integrated into the full pipeline. For the latter, parts of the image which hand-crafted detectors miss are simply discarded for training.

In this paper, we propose a sparse-matching method with a novel deep architecture, which we name LF-Net, for Local Feature Network, that is trainable end-to-end *and* does not require using a hand-crafted detector to generate training data. Instead, we use image pairs for which we know the relative pose and corresponding depth maps, which can be obtained either with laser scanners or shape-from-structure algorithms [34], without *any* further annotation.

Being thus given dense correspondence data, we could train a feature extraction pipeline by selecting a number of keypoints over two images, computing descriptors for each keypoint, using the ground truth to determine which ones match correctly across images, and use those to learn good descriptors. This is, however, not feasible in practice. First, extracting multiple maxima from a score map is inherently not differentiable. Second, performing this operation over each image produces two

disjoint sets of keypoints which will typically produce very few ground truth matches, which we need to train the descriptor network, and in turn guide the detector towards keypoints which are distinctive and good for matching.

We therefore propose to create a virtual target response for the network, using the ground-truth geometry in a non-differentiable way. Specifically, we run our detector on the first image, find the maxima, and then optimize the weights so that when run on the second image it produces a *clean response map* with sharp maxima at the right locations. Moreover, we warp the keypoints selected in this manner to the other image using the ground truth, guaranteeing a large pool of *ground truth matches*. Note that while we break differentiability in one branch, the other one can be trained end to end, which lets us learn discriminative features by learning the entire pipeline at once. We show that our method greatly outperforms the state-of-the-art.

## 2 Related work

Since the appearance of SIFT [23], local features have played a crucial role in computer vision, becoming the *de facto* standard for wide-baseline image matching [14]. They are versatile [23, 29, 47] and remain useful in many scenarios. This remains true even in competition with deep network alternatives, which typically involve dense matching [9, 43, 45, 52, 53] and tend to work best on narrow baselines, as they can suffer from occlusions, which local features are robust against.

Typically, feature extraction and matching comprises three stages: finding interest points, estimating their orientation, and creating a descriptor for each. SIFT [23], along with more recent methods [1, 5, 32, 48] implements the entire pipeline. However, many other approaches target some of their individual components, be it feature point extraction [31, 44], orientation estimation [50], or descriptor generation [36, 40, 41]. One problem with this approach is that increasing the performance of one component does not necessarily translate into overall improvements [35, 48].

Next, we briefly introduce some representative algorithms below, separating those that rely on hand-crafted features from those that use Machine Learning techniques extensively.

**Hand-crafted.** SIFT [23] was the first widely successful attempt at designing an integrated solution for local feature extraction. Many subsequent efforts focused on reducing its computational requirements. For instance, SURF [5] used Haar filters and integral images for fast keypoint detection and descriptor extraction. DAISY [41] computed dense descriptors efficiently from convolutions of oriented gradient maps. The literature on this topic is very extensive—we refer the reader to [28].

**Learned.** While methods such as FAST [30] used machine learning techniques to extract keypoints, most early efforts in this area targeted descriptors, *e.g.* using metric learning [38] or convex optimization [37]. However, with the advent of deep learning, there has been a renewed push towards replacing all the components of the standard pipeline by convolutional neural networks.

– *Keypoints.* In [44], piecewise-linear convolutional filters were used to make keypoint detection robust to severe lighting changes. In [33], neural networks are trained to rank keypoints. The latter is relevant to our work because no annotations are required to train the keypoint detector, but both methods are optimized for repeatability and not for the quality of the associated descriptors. Deep networks have also been used to learn covariant feature detectors, particularly towards invariance against affine transformations due to viewpoint changes [22, 26].

– *Orientations.* The method of [50] is the only one we know of that focuses on improving orientation estimates. It uses a siamese network to predict the orientations that minimize the distance between the orientation-dependent descriptors of matching keypoints, assuming that the keypoints have been extracted using some other technique.

– *Descriptors.* The bulk of methods focus on descriptors. In [13, 51], the comparison metric is learned by training Siamese networks. Later works, starting with [36], rely on hard sample mining for training and the $l_2$ norm for comparisons. A triplet-based loss function was introduced in [3], and in [25], negative samples are mined over the entire training batch. More recent efforts further increased performance using spectral pooling [46] and novel loss formulations [19]. However, none of these take into account what kind of keypoint they are working and typically use only SIFT.

Crucially, performance improvements in popular benchmarks for a single one of either of these three components do not always survive when evaluating the whole pipeline [35, 48]. For example,

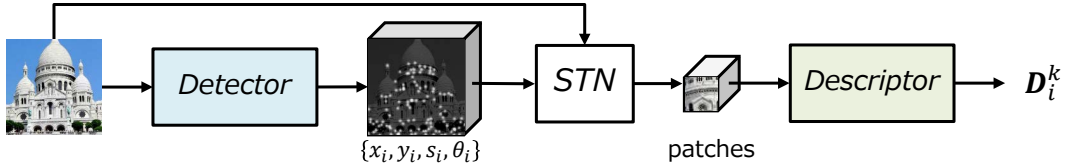

(a) The LF-Net architecture. The *detector* network generates a scale-space score map along with dense orientation estimates, which are used to select the keypoints. Image patches around the chosen keypoints are cropped with a differentiable sampler (STN) and fed to the *descriptor* network, which generates a descriptor for each patch.

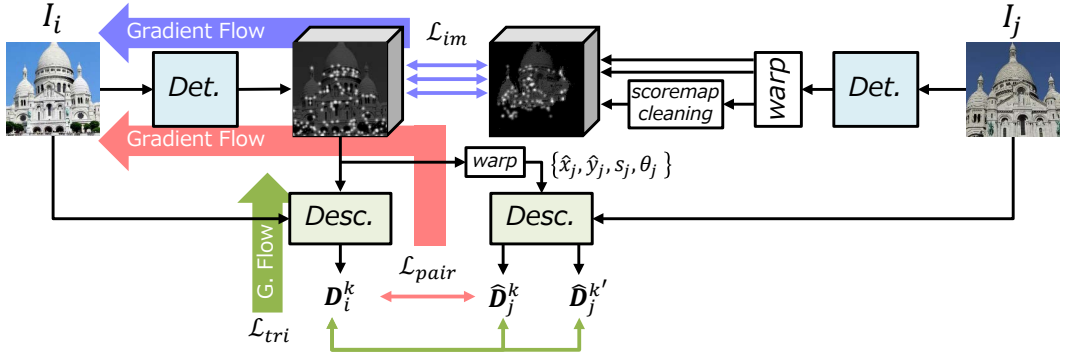

(b) For training we use a *two-branch* LF-Net, containing two identical copies of the network, processing two corresponding images $I_i$ and $I_j$. Branch $j$ (right) is used to generate a supervision signal for branch $i$ (left), created by warping the results from $i$ to $j$. As this is not differentiable, we optimize only over branch $i$, and update the network copy for branch $j$ in the next iteration. We omit the samplers in this figure, for simplicity.

Figure 1: (a) The Local Feature Network (LF-Net). (b) Training with two LF-Nets.

keypoints are often evaluated on repeatability, which can be misleading because they may be repeatable but useless for matching purposes. Descriptors can prove very robust against photometric and geometric transformations, but this may be unnecessary or even counterproductive when patches are well-aligned, and results on the most common benchmark [7] are heavily saturated.

This was demonstrated in [48], which integrated previous efforts [36, 44, 50] into a fully-differentiable architecture, reformulating the entire keypoint extraction pipeline with deep networks. It showed that not only is joint training necessary for optimal performance, but also that standard SIFT still outperforms many modern baselines. However, their approach still relies on SIFT keypoints for training, and as a result it can not learn where SIFT itself fails. Along the same lines, a deep network was introduced in [11] to match images with a keypoint-based formulation, assuming a homography model. However, it was largely trained on synthetic images or real images with affine transformations, and its effectiveness on practical wide-baseline stereo problems remains unproven.

## 3 Method

Fig. 1 depicts the LF-Net architecture (top), and our training pipeline with two LF-Nets (bottom). In the following we first describe our network in Section 3.1. We break it down into its individual components and detail how they are connected in order to build a complete feature extraction pipeline. In Section 3.2, we introduce our training architecture, which is based on two LF-Net copies processing separate images with non-differentiable components, along with the loss function used to learn the weights. In Section 3.3 we outline some technical details.

### 3.1 LF-Net: a Local Feature Network

LF-Net has two main components. The first one is a dense, multi-scale, fully convolutional network that returns keypoint locations, scales, and orientations. It is designed to achieve fast inference time, and to be agnostic to image size. The second is a network that outputs local descriptors given patches cropped around the keypoints produced by the first network. We call them *detector* and *descriptor*.

In the remainder of this section, we assume that the images have been undistorted using the camera calibration data. We convert them to grayscale for simplicity and simply normalize them individually using their mean and standard deviation [42]. As will be discussed in Section 4.1, depth maps and camera parameters can all be obtained using off-the-shelf SfM algorithms [34]. As depth measurements are often missing around 3D object boundaries—especially when computed SfM algorithms—image regions for which we do not have depth measurements are masked and discarded during training.

**Feature map generation.** We first use a fully convolutional network to generate a rich feature map $\mathbf{o}$ from an image $\mathbf{I}$, which can be used to extract keypoint locations as well as their attributes, *i.e.*, scale and orientation. We do this for two reasons. First, it has been shown that using such a mid-level representation to estimate multiple quantities helps increase the predictive power of deep nets [21]. Second, it allows for larger batch sizes, that is, using more images simultaneously, which is key to training a robust detector.

In practice, we use a simple ResNet [15] layout with three blocks. Each block contains $5 \times 5$ convolutional filters followed by batch normalization [17], leaky-ReLU activations, and another set of $5 \times 5$ convolutions. All convolutions are zero-padded to have the same output size as the input, and have 16 output channels. In our experiments, this has proved more successful that more recent architectures relying on strided convolutions and pixel shuffling [11].

**Scale-invariant keypoint detection.** To detect scale-invariant keypoints we propose a novel approach to scale-space detection that relies on the feature map $\mathbf{o}$. To generate a scale-space response, we resize it $N$ times, at uniform intervals between $1/R$ and $R$, where $N = 5$ and $R = \sqrt{2}$ in our experiments. These are convolved with $N$ independent $5 \times 5$ filters size, which results in $N$ score maps $\mathbf{h}^n$ for $1 \leq n < N$, one for each scale. To increase the saliency of keypoints, we perform a differentiable form of non-maximum suppression by applying a softmax operator over $15 \times 15$ windows in a convolutional manner, which results in $N$ sharper score maps, $\hat{\mathbf{h}}^n_{1 \leq n < N}$. Since the non-maximum suppression results are scale-dependent, we resize each $\hat{\mathbf{h}}^n$ back to the original image size, which yields $\bar{\mathbf{h}}^n_{1 \leq n < N}$. Finally, we merge all the $\bar{\mathbf{h}}^n$ into a final scale-space score map, $\mathbf{S}$, with a softmax-like operation. We define it as

$$\mathbf{S} = \sum_n \bar{\mathbf{h}}^n \odot \mathrm{softmax}_n\left(\bar{\mathbf{h}}^n\right) \quad , \tag{1}$$

where $\odot$ is the Hadamard product.

From this scale-invariant map we choose the top $K$ pixels as keypoints, and further apply a local softargmax [8] for sub-pixel accuracy. While selecting the the top $K$ keypoints is not differentiable, this does not stop gradients from back-propagating through the selected points. Furthermore, the sub-pixel refinement through softargmax also makes it possible for gradients to flow through with respect to keypoint coordinates. To predict the scale at each keypoint, we simply apply a softargmax operation over the scale dimension of $\bar{\mathbf{h}}^n$. A simpler alternative would have been to directly regress the scale once a keypoint has been detected. However, this turned out to be less effective in practice.

**Orientation estimation.** To learn orientations we follow the approach of [48, 50], but on the shared feature representation $\mathbf{o}$ instead of the image. We apply a single $5 \times 5$ convolution on $\mathbf{o}$ which outputs two values for each pixel. They are taken to be the sine and cosine of the orientation and and used to compute a dense orientation map $\boldsymbol{\theta}$ using the `arctan` function.

**Descriptor extraction.** As discussed above, we extract from the score map $\mathbf{S}$ the $K$ highest scoring feature points and their image locations. With the scale map $\mathbf{s}$ and orientation map $\boldsymbol{\theta}$, this gives us $K$ quadruplets of the form $\mathbf{p}^k = \{x, y, s, \theta\}^k$, for which we want to compute descriptors.

To this end, we consider image patches around the selected keypoint locations. We crop them from the normalized images and resize them to $32 \times 32$. To preserve differentiability, we use the bilinear sampling scheme of [18] for cropping. Our descriptor network comprises three $3 \times 3$ convolutional filters with a stride of 2 and 64, 128, and 256 channels respectively. Each one is followed by batch normalization and a ReLU activation. After the convolutional layers, we have a fully-connected 512-channel layer, followed by batch normalization, ReLU, and a final fully-connected layer to reduce the dimensionality to $M=256$. The descriptors are $l_2$ normalized and we denote them as $\mathbf{D}$.

## 3.2 Learning LF-Net

As shown in Fig. 1, we formulate the learning problem in terms of a two-branch architecture which takes as input two images of the same scene, $\mathbf{I}_i$ and $\mathbf{I}_j$, $i \neq j$, along with their respective depth maps and the camera intrinsics and extrinsics, which can be obtained from conventional SfM methods. Given this data, we can warp the score maps to determine ground truth correspondences between images. One distinctive characteristic of our setup is that branch $j$ holds the components which break differentiability, and is thus never back-propagated, in contrast to a traditional Siamese architecture. To do this in a mathematically sound way, we take inspiration from Q-learning [27] and use the parameters of the previous iteration of the network for this branch.

We formulate our training objective as a combination of two types of loss functions: image-level and patch-level. Keypoint detection requires image-level operations and also affects where patches are extracted, thus we use both image-level and patch-level losses. For the descriptor network, we use only patch-level losses as they operate independently for each patch once keypoints are selected.

**Image-level loss.** We can warp a score map with rigid-body transforms [12], using the projective camera model. We call this the SE(3) module $w$, which in addition to the score maps takes as input the camera pose $\mathbf{P}$, calibration matrix $\mathbf{K}$ and depth map $\mathbf{Z}$, for both images—note that we omit the latter three for brevity. We propose to select $K$ keypoints from the warped score map for $\mathbf{I}_j$ with standard, non-differentiable non-maximum suppression, and generate a clean score map by placing Gaussian kernels with standard deviation $\sigma = 0.5$ at those locations. We denote this operation $g$. Note that while it is non-differentiable, it only takes place on branch $j$, and thus has no effect in the optimization. Mathematically, we write

$$\mathcal{L}_{im}(\mathbf{S}_i, \mathbf{S}_j) = |\mathbf{S}_i - g(w(\mathbf{S}_j))|^2 \ . \tag{2}$$

Here, as mentioned before, occluded image regions are not used for optimization.

**Patch-wise loss.** With existing methods [3, 25, 36], the pool of pair-wise relationships is predefined before training, assuming a detector is given. More importantly, forming these pairs from two disconnected sets of keypoints will produce too many outliers for the training to ever converge. Finally, we want the gradients to flow back to the keypoint detector network, so that we are able to learn keypoints that are good for matching.

We propose to solve this problem by leveraging the ground truth camera motion and depth to form sparse patch correspondences on the fly, by warping the detected keypoints. Note that we are only able to do this as we warp over branch $j$ and back-propagate through branch $i$.

More specifically, once $K$ keypoints are selected from $\mathbf{I}_i$, we warp their spatial coordinates to $\mathbf{I}_j$, similarly as we do for the score maps to compute the image-level loss, but in the opposite direction. Note that we form the keypoint with scale and orientation from branch $j$, as they are not as sensitive as the location, and we empirically found that it helps the optimization. We then extract descriptors at these corresponding regions $\mathbf{p_i^k}$ and $\hat{\mathbf{p}}_\mathbf{j}^\mathbf{k}$. If a keypoint falls on occluded regions after warping, we drop it from the optimisation process. With these corresponding regions and their associated decriptors $\mathbf{D}_i^k$ and $\hat{\mathbf{D}}_j^k$ we form $\mathcal{L}_{pair}$ which is used to train the detector network, *i.e.*, the keypoint, orientation, and scale components. Mathematically we write

$$\mathcal{L}_{pair}(\mathbf{D}_i^k, \hat{\mathbf{D}}_j^k) = \sum_k |\mathbf{D}_i^k - \hat{\mathbf{D}}_j^k|^2 \ . \tag{3}$$

Similarly, in addition to the descriptors, we also enforce geometrical consistency over the orientation of the detected and warped points. We thus write

$$\mathcal{L}_{geom}(s_i^k, \theta_i^k, \hat{s}_i^k, \hat{\theta}_j^k) = \lambda_{ori} \sum_k |\theta_i^k - \hat{\theta}_j^k|^2 + \lambda_{scale} \sum_k |s_i^k - \hat{s}_j^k|^2 \ , \tag{4}$$

where $\hat{s}$ and $\hat{\theta}$ respectively denotes the warped scale and orientation of a keypoint, by using the relative camera pose between the two images, and $\lambda_{ori}$ and $\lambda_{scale}$ are weights.

**Triplet loss for descriptors.** To learn the descriptor, we also need to consider non-corresponding pairs of patches. Similar to [3], we form a triplet loss to learn the ideal embedding space for the

patches. However, for the positive pair we use the ground-truth geometry to find a match, as described above. For the negative—non-matching—pairs, we employ a progressive mining strategy to obtain the most informative patches possible. Specifically, we sort the negatives for each sample by loss in decreasing order and sample randomly over the top $M$, where $M = \max(5, 64e^{\frac{0.6k}{1000}})$, where $k$ is the current iteration, *i.e.*, we start with a pool of the 64 hardest samples and reduce it as the networks converge, up to a minimum of 5. Sampling informative patches is critical to learn discriminative descriptors, and random sampling will provide too many easy negative samples.

With the matching and non-matching pairs, we form the triplet loss as:

$$\mathcal{L}_{tri}(\mathbf{D}_i^k, \hat{\mathbf{D}}_j^k, \hat{\mathbf{D}}_j^{k'}) = \sum_k \max\left(0, |\mathbf{D}_i^k - \hat{\mathbf{D}}_j^k|^2 - |\mathbf{D}_i^k - \hat{\mathbf{D}}_j^{k'}|^2 + C\right) . \tag{5}$$

where $k' \neq k$, *i.e.*, it can be any non-corresponding sample, and $C$=1 is the margin.

**Loss function for each sub-network.** In summary, the loss function that is used to learn each sub-network is the following:

- Detector loss: $\mathcal{L}_{det} = \mathcal{L}_{im} + \lambda_{pair}\mathcal{L}_{pair} + \mathcal{L}_{geom}$
- Descriptor loss: $\mathcal{L}_{desc} = \mathcal{L}_{tri}$

### 3.3 Technical details, otimization, and inference

To make the optimization more stable, we flip the images on each branch and merge the gradients before updating. We emphasize here that with our loss, the gradients for the patch-wise loss can safely back-propagate through branch $i$, including the top $K$ selection, to the image-level networks. Likewise, the softargmax operator used for keypoint extraction allows the optimization to differentiate the patch-wise loss with respect to the location of the keypoints.

Note that for inference we keep a single copy of the network, *i.e.*, the architecture of Fig. 1-(a), and simply run the differentiable part of the framework, branch $i$. Although differentiability is no longer a concern, we still rely, for simplicity, on the spatial SoftMax for non-maximum supression and the softargmax and spatial transformers for patch sampling. Even so, our implementation can extract 512 keypoints from QVGA frames (320×240) at 62 fps and from VGA frames (640×480) at 25 fps (42 and 20 respectively for 1024 keypoints), on a Titan X PASCAL. Please refer to the supplementary material for a thorough comparison of computational costs.

While training we extract 512 keypoints, as larger numbers become problematic due to memory constraints. This also allows us to maintain a batch with multiple image pairs (6), which helps convergence. Note that at test time we can choose as many keypoints as desired. As datasets with natural images are composed of mostly upright images and are thus rather biased in terms of orientation, we perform data augmentation by randomly rotating the input patches by $\pm180^o$, and transform the camera's roll angle accordingly. We also perform scale augmentation by resizing the input patches by $1/\sqrt{2}$ to $\sqrt{2}$, and transforming the focal length accordingly. This allows us to train models comparable to traditional keypoint extraction pipelines, *i.e.*, with built-in invariance to rotation and scaling. However, in practice, many of the images in our indoors and outdoors examples are upright, and the best-performing models are obtained by disabling these augmentations as well as the orientation ans scale estimation completely. This is the approach followed by learned SLAM front-ends such as [11]. We consider both strategies in the next section.

For optimization, we use ADAM [20] with a learning rate of $10^{-3}$. To balance the loss function for the detector network we use $\lambda_{pair} = 0.01$, and $\lambda_{ori} = \lambda_{scale} = 0.1$. Our implementation is written in TensorFlow and is publicly available.[1]

## 4 Experiments

### 4.1 Datasets

We consider both indoors and outdoors images as their characteristics drastically differ, as shown in Fig. 2. For indoors data we rely on ScanNet [10], an RGB-D dataset with over 2.5M images,

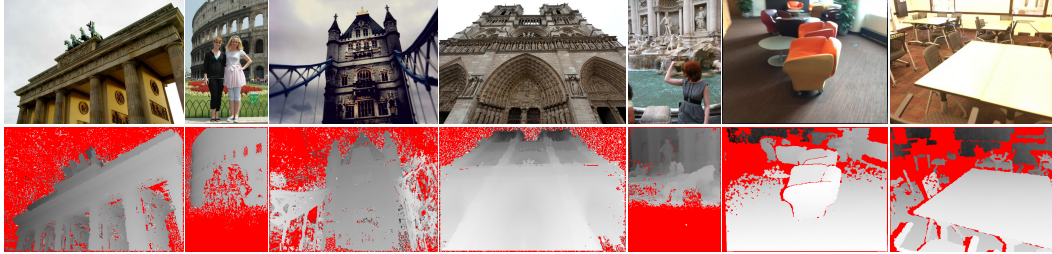

Figure 2: Samples from our indoors and outdoors datasets. Image regions without depth measurements, due to occlusions or sensor shortcomings, are drawn in red, and are simply excluded from the optimization. Note the remaining artefacts in the depth maps for outdoors images.

including accurate camera poses from SfM reconstructions. These sequences show office settings with specularities and very significant blurring artefacts, and the depth maps are incomplete due to sensing failures, specially around 3D object boundaries. The dataset provides training, validation, and test splits that we use accordingly. As this dataset is very large, we only use roughly half of the available sequences for training and validation, but test on the entire set of 312 sequences, pre-selected by the authors, with the exception of LIFT. For LIFT we use a random subset of 43 sequences as the authors' implementation is too slow. To prevent selecting pairs of images that do not share any field of view, we sample images 15 frames away, guaranteeing enough scene overlap. At test time, we consider multiple values for the frame difference to evaluate increasing baselines.

For outdoors data we use 25 photo-tourism image collections of popular landmarks collected by [16, 39]. We run COLMAP [34] to obtain dense 3D reconstructions, including dense but noisy and inaccurate depth maps for every image. We post-process the depth maps by projecting each image pixel to 3D space at the estimated depth, and mark it as invalid if the closest 3D point from the reconstruction is further than a threshold. The resulting depth maps are still noisy, but many occluded pixels are filtered out as shown in Fig. 2. To guarantee a reasonable degree of overlap for each image pair we perform a visibility check using the SfM points visible over both images. We consider bounding boxes twice the size of those containing these points to extract image regions roughly corresponding, while ignoring very small ones. We use 14 sequences for training and validation, splitting the images into training and validation subsets by with a 70:30 ratio, and sample up to $50k$ pairs from each different scene. For testing we use the remaining 11 sequences, which were not used for training or validation, and sample up to $1k$ pairs from each set. We use square patches size $256 \times 256$ for training, for either data type.

## 4.2 Baselines and metrics

We consider the following full local feature pipelines: SIFT [23], SURF [6] ORB [32], A-KAZE [2], LIFT [48], and SuperPoint [11], using the authors' release for the learned variants, LIFT and SuperPoint, and OpenCV for the rest. For ScanNet, we test on 320×240 images, which is commensurate with the patches cropped while training. We do the same for the baselines, as their performance seems to be better than at higher resolutions, probably due to the low-texture nature of the images. For the outdoors dataset, we resize the images so that the largest dimensions is 640 pixels, as they are richer in texture, and all methods work better at this resolution. Similarly, we extract 1024 keypoints for outdoors images, but limit them to 512 for Scannet, as the latter contains very little texture.

To evaluate the entire local feature pipeline performance, we use the matching score [24], which is defined as the ratio of estimated correspondences that are correct according to the ground-truth geometry, after obtaining them through nearest neighbour matching with the descriptors. As our data exhibits complex geometry, and to emphasize accurate localization of keypoints, similar to [31] we use a 5-pixel threshold instead of the overlap measure used in [24]. For results under different thresholds please refer to the supplementary material.

## 4.3 Results on outdoors data

For this experiment we provide results independently for each sequence, in addition to the average. Due to the nature of the data, results vary from sequence to sequence. We provide quantitative results

Table 1: Matching score for the outdoors dataset. Best results are marked in bold.

| Sequence | SIFT | SURF | A-KAZE | ORB | LIFT | SuperPoint | LF-Net w/rot-scl | LF-Net w/o rot-scl |
|---|---|---|---|---|---|---|---|---|
| 'british_museum' | .265 | .288 | .287 | .055 | .318 | .468 | .456 | **.560** |
| 'florence_cathedral_side' | .181 | .158 | .116 | .027 | .204 | .359 | .285 | **.362** |
| 'lincoln_memorial_statue' | .193 | .204 | .167 | .037 | .220 | **.384** | .288 | .357 |
| 'london_bridge' | .177 | .170 | .168 | .057 | .250 | **.468** | .342 | .452 |
| 'milan_cathedral' | .188 | .221 | .194 | .021 | .237 | .401 | .423 | **.520** |
| 'mount_rushmore' | .225 | .241 | .210 | .041 | .300 | .512 | .379 | **.543** |
| 'piazza_san_marco' | .115 | .115 | .106 | .026 | .145 | .253 | .233 | **.287** |
| 'reichstag' | .212 | .209 | .175 | .097 | .246 | .414 | .379 | **.466** |
| 'sagrada_familia' | .199 | .175 | .140 | .031 | .205 | .295 | .311 | **.341** |
| 'st_pauls_cathedral' | .149 | .160 | .150 | .026 | .177 | .319 | .266 | **.347** |
| 'united_states_capitol' | .118 | .103 | .086 | .028 | .134 | .220 | .173 | **.232** |
| Average | .184 | .186 | .164 | .041 | .221 | .372 | .321 | **.406** |

Table 2: Matching score for the indoors dataset. Best results are marked in bold.

| Frame difference | SIFT | SURF | A-KAZE | ORB | LIFT | SuperPoint | LF-Net (w/rot-scl) | LF-Net (w/o rot-scl) |
|---|---|---|---|---|---|---|---|---|
| 10 | .320 | .464 | .465 | .223 | .389 | **.688** | .607 | **.688** |
| 20 | .264 | .357 | .337 | .172 | .283 | **.599** | .497 | .574 |
| 30 | .226 | .290 | .260 | .141 | .247 | **.525** | .419 | .483 |
| 60 | .152 | .179 | .145 | .089 | .147 | **.358** | .276 | .300 |
| Average | .241 | .323 | .302 | .156 | .267 | **.542** | .450 | .511 |

in Table 1 and qualitative examples in Fig. 3. As previously noted in Section 3.3, most images are upright and at similar scales, so that the best results are obtained simply bypassing scale and rotation estimation. In order to compare with SuperPoint, we train our models in this setup ('w/o rot-scl'). LF-Net gives best performance and outperforms SuperPoint by 9% relative. In order to compare with traditional pipelines with explicit rotation and scale estimation, we also train our models with the augmentations described in section Section 3.3 ('w/ rot-scl'). Out of these baselines, our approach outperforms the closest competitor, LIFT, by 45% relative. We provide qualitative examples in Fig. 3.

## 4.4 Results on indoors data

This dataset contains video sequences. To evaluate performance over different baselines, we sample image pairs at different frame difference values: 10, 20, 30, and 60. At 10 the images are very similar, whereas at 60 there is a significant degree of camera motion—note that our method is trained exclusively at a 15-frame difference. Results are shown in Table 2. As for the outdoors case, the images in this dataset are mostly upright, so we report results for models trained with and without explicit rotation and scale detection. LF-Net achieves the same performance as SuperPoint on the 10-frame difference case, but performs worse for larger frame differences—6% relative on average. We believe this to be due to the fact that in office-like indoor settings with little texture such as the ones common in this dataset, 3D object boundaries are often the most informative features, and they may be in many cases excluded from LF-Net training due to depth inaccuracies from the Kinect sensor. Note however that the self-supervised training strategy used by SuperPoint can also be applied to LF-Net to boost performance. Finally, among pipelines with explicit rotation and scale estimation, LF-Net outperforms the closest competitor, SURF, by 39% relative.

## 4.5 Ablation study

As demonstrated in [48], it is crucial to train the different components jointly when learning a feature extraction pipeline. As an ablation study, we consider the case where $\lambda_{pair} = 0$, *i.e.*, we do not train the detector with the patch-wise loss, effectively separating the training of the detector and the

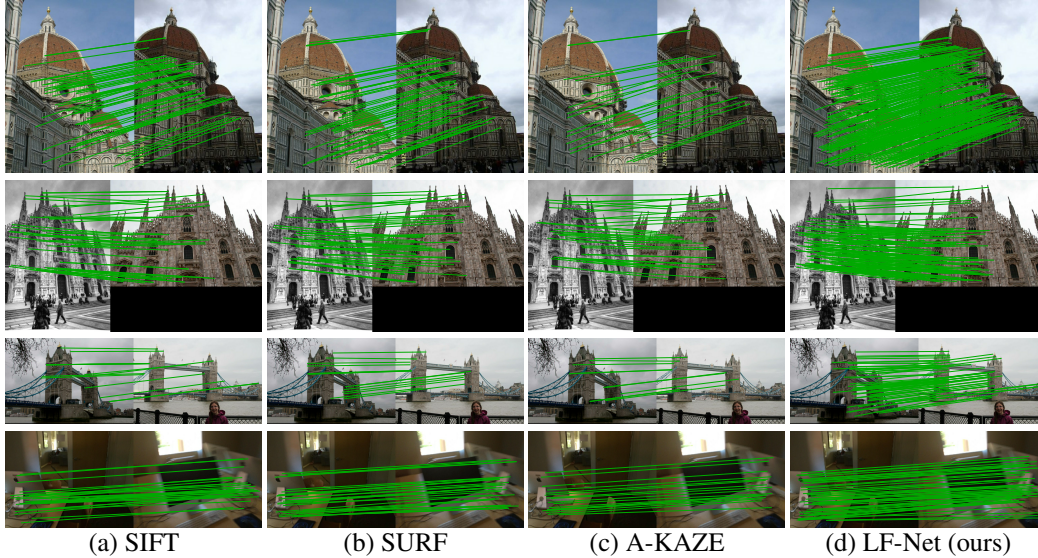

| (a) SIFT | (b) SURF | (c) A-KAZE | (d) LF-Net (ours) |

Figure 3: Qualitative matching results, with correct matches drawn in green.

descriptor. For this experiment we consider models trained with rotation and scale augmentations. While they still produce state-of-the-art results w.r.t. the rotation-sensitive baselines, they tend to converge earlier, and training them jointly increases average matching score by 7% relative for the outdoors dataset (.299 to .321) and 1% relative for the indoors dataset (.445 to .450). Again, we believe that the small performance increase on indoors data is due to the inherent limits of the device used to capture the depth estimates. For more detailed results please refer to the supplementary appendix.

### 4.6 Additional results

More experimental results that could not fit in the paper due to spatial constraints are available as a supplementary appendix: a generalization study, training and testing on very different datasets; a performance evaluation for different pixel thresholds, to evaluate the precision of the detected features; results over the 'hpatches' dataset [4]; a study to benchmark performance under orientation changes; detailed results for the ablation experiments of Section 4.5; and computational cost estimates.

## 5 Conclusions

We have proposed LF-Net, a novel deep architecture to learn local features. It embeds the entire feature extraction pipeline, and can be trained end-to-end with just a collection of images. To allow training from scratch without hand-crafted priors, we devise a two-branch setup and create virtual target responses iteratively. We run this non-differentiable process in one branch while optimizing over the other, which we keep differentiable, and show they converge to an optimal solution. Our method outperforms the state of the art by a large margin, on both indoor and outdoor datasets, at 60 fps for QVGA images.

### Acknowledgments

This work was partially supported by the Natural Sciences and Engineering Research Council of Canada (NSERC) Discovery Grant "Deep Visual Geometry Machines" (RGPIN-2018-03788), and by systems supplied by Compute Canada.

## Footnotes

[1] https://github.com/vcg-uvic/lf-net-release

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
