[Supplementary Material]

# 6 Appendix

## 6.1 Generalization performance

We test the models trained on indoors data with outdoors data, and vice-versa, to benchmark their generalization performance, determining how well they perform on images with very different geometric and photometric properties than seen while training. For this experiment we use models trained with rotation and scale augmentations. The performance drop training with outdoors data and testing with indoors data is 18% relative (.450 to .370 matching score). The performance drop training with indoors data and testing with outdoors data is 8% relative (.321 to .295 in matching score). Compared to the rotation-sensitive baselines, the performance still remains state-of-the-art, by a large margin.

## 6.2 Computational cost

As outlined in Section 3.3, LF-Net can extract 512 keypoints from QVGA frames (320×240) at 62 fps and from VGA frames (640×480) at 25 fps (42 and 20 respectively for 1024 keypoints), on a Titan X PASCAL. Table 3 lists the computational cost for different keypoint extraction pipelines. As they are designed to operate in different regimes, we normalize the timings by the number of keypoints generated by each method, and provide the mean cost of extracting a single keypoint in micro-seconds. For SuperPoint we use the available implementation, which has some computational overhead due to image loading and pre-processing—the authors claim a runtime of 70 fps at VGA resolutions, which should be achievable. LIFT's modular implementation is provided only for reference, and should be highly optimizable at inference time.

Table 3: Computational cost for different keypoint extraction pipelines. The estimates are normalized by the number of keypoints and given in micro-seconds. *) For SuperPoint, running on smaller image size gives less keypoints, and therefore the increase in per-keypoint computation time.

| Input size | $320 \times 240$ | $640 \times 480$ |
|---|---|---|
| LF-Net (512 kp) | 31.5 | 78.1 |
| LF-Net (1024 kp) | 23.2 | 48.8 |
| SIFT | 62.7 | 55.9 |
| SURF | 10.0 | 10.0 |
| A-KAZE | 39.1 | 25.1 |
| ORB | 17.6 | 49.3 |
| LIFT | $509 \cdot 10^3$ | $119 \cdot 10^3$ |
| SuperPoint | $263.7^*$ | $130.9^*$ |

## 6.3 Geometric precision

In the paper we consider only a threshold of 5 pixels to determine correct matches, which does not necessarily capture how accurate each keypoint is. To overcome the limitation of this setup, in Tables 4 and 5 we report matching score results for different pixel thresholds, from 1 to 5. Our method outperforms SIFT and SURF at any threshold other than sub-pixel, and can double their performance at higher thresholds for certain datasets. Note that sub-pixel results are not fully trustworthy, as the depth estimates are quite noisy, and our method would benefit from better training and test data.

Table 4: Matching score: Outdoors

| Threshold (pix) | 1 | 2 | 3 | 4 | 5 |
|---|---|---|---|---|---|
| SIFT | **.111** | .162 | .175 | .180 | .184 |
| SURF | .068 | .130 | .160 | .176 | .186 |
| LF-Net | .102 | **.229** | **.284** | **.308** | **.321** |

Table 5: Matching score: Indoors. Please note that these results were obtained over a subset of the data used in Section 4.4, which explains the small difference with Table 2.

| Threshold (pix) | 1 | 2 | 3 | 4 | 5 |
|---|---|---|---|---|---|
| SIFT | .072 | .151 | .196 | .223 | .241 |
| SURF | **.074** | .170 | .240 | .288 | .323 |
| LF-Net | .066 | **.189** | **.298** | **.383** | **.450** |

## 6.4 Results on the 'hpatches' dataset

In Table 6 we report matching score results for the 'hpatches' dataset [4]. As in the previous section, we consider multiple pixel thresholds. As before, LF-Net outperforms classical algorithms except at the sub-pixel level.

Table 6: Matching score: Outdoors

| Feature | Sequence | Pixel threshold | | | | |
|---|---|---|---|---|---|---|
| | | 1 | 2 | 3 | 4 | 5 |
| SIFT | | .206 | .290 | .324 | .310 | .319 |
| SURF | Viewpoint | .138 | .258 | .312 | .337 | .351 |
| LF-Net | | .131 | .265 | .311 | .329 | .339 |
| SIFT | | .180 | .248 | .267 | .274 | .279 |
| SURF | Illumination | .162 | .245 | .309 | .281 | .298 |
| LF-Net | | .171 | .310 | .360 | .378 | .389 |
| SIFT | | **.193** | .269 | .288 | .296 | .301 |
| SURF | Average | .150 | .252 | .296 | .317 | .330 |
| LF-Net | | .151 | **.288** | **.335** | **.354** | **.364** |

## 6.5 Rotation invariance

To measure the rotation invariance of each keypoint type, we match a subset of the outdoors datasets, 500 image pairs from different sequences, while applying in-plane rotations to the images every $10^o$, from 0 to $360^o$, and average the matching score over each bin. The results are listed in Table 7. As expected, the models learned without rotation invariance may excel when the images are well aligned (Table 1) but perform very poorly overall.

Table 7: Average matching score under rotations

| Feature | Avg. matching score |
|---|---|
| SIFT | .265 |
| SURF | .208 |
| LF-Net (w/ rot-scl) | **.338** |
| SuperPoint | .092 |
| LF-Net (w/o rot-scl) | .074 |