[Reviews · NeurIPS 2018]

Reviewer 1



Authors introduce a novel method for learning local feature detector and descriptors in a single framework. Compared to previous works, this method uses pair of images (instead of pair of image patches) and employs data with estimated depth maps and camera parameters. Learning on full images is achieved with an image loss which bypasses the lack of differentiability by computing gradient for only one branch. Even though the method is novel, I have concerns regarding the clarity and correctness of the evaluation. The main strength of this work is the fact that it uses whole images for training the local feature detector. Thanks to the trick of using a global feature representation, authors are able to train with multiple images at once which mitigates the issue of an SGD update fitting a single image. Another claim, that the paper is able to discover valid points from scratch is unclear as it uses SFM results which, up to certain extent depend on quality of SIFT detector and descriptors. This makes their claim weaker and comparable to [1] which as well does not use an external local feature detector to bootstrap the training (and authors forget to cite this work). However, my main concern is the lack of detail in the evaluation. As shown in many previous works, for evaluation of local feature matching algorithms, the measurement region size is a crucial parameter for a descriptor performance. Thus it is important to make sure that similar measurement regions size is used for all evaluated baselines in order to compare them fairly. A common approach is to evaluate the descriptor separately on pre-extracted patches [2, 3] which makes sure that the descriptors have access to the same information. And not all datasets have "heavily saturated scores". In this regard, it is not clear whether this has been taken into consideration. Similarly, it is not clear whether a similar amount of detected points was considered for each algorithm, as this can influence both the qualitative and quantitative evaluation. However, a curse of local features is that each manuscript uses a different evaluation, thus a direct comparison is difficult. Separate evaluation of the detector a descriptors can yield some insight into the source of the main improvements, as e.g. the geometric precision of the detector is important for SFM application etc. Additionally, a speed comparison to other algorithms is missing. Comparison of performance on existing datasets would also be useful (such as in [4]), as it would allow to directly compare the numbers for the investigated baselines. The writing does not seem to be well organized, mainly the Method section. The Figure 1 is hard to follow and explanation of branch i and j, used in section 3.2 is described in section 3.3. Similarly, it is not clear how the scores are combined across scales (Equation 1, across what elements is the sofmtax applied?), or how is scale taken into consideration while generating the "clean score maps", or how many points are considered for g. Similarly, it is not clear how the "split" mode of training works, considering that the orientation estimation is not being trained (line 203 and 273). It is not clear whether the common feature map is pre-trained or trained from scratch and how the camera parameters are used in the CNN (probably issue of wording). Even though the method is quite interesting, my decision is mainly based on the issues with the evaluation which does not allow to verify fair comparison to existing methods. Additionally, I feel this submission would be more apt for computer vision conferences. [1] Learning covariant feature detectors. K Lenc, A Vedaldi - Workshop of European Conference on Computer Vision, 2016. [2] Learning local image descriptors. S Winder and M Brown, CVPR 2007. [3] HPatches: A benchmark and evaluation of handcrafted and learned local descriptors, Balntas et al., CVPR 2017 [4] LIFT: Learned Invariant Feature Transform, Yi et al, ECCV 2016. -------------------- Rebuttal feedback I feel that the authors have addressed my main concerns really well and I am willing to improve the "overall score". It seems that the additional results provided in the rebuttal give additional evidence that the method is superior to existing ones and thus I would recommend the authors to include those in the final version of the paper together with improving the clarity of the text. I am still slightly skeptical about their definition of the support region, provided as an average size of the region in pixels, which is a bit unconventional but ultimately seems to make sense. Traditionally, one uses magnification factor, however this actually addresses the issue of a possible bias in the detected scales.

Reviewer 2



Overall this is significant and original work because it is one of the first works to formulate keypoint detection and description as an end-to-end learning problem without relying on the output of hand-crafted detector for supervision. The approach follows the traditional SIFT recipe closely, replacing each component with a learning based counterpart, much like in LIFT. The authors side-step the non-differentiability of the Non-Max-Suppression step by setting up a siamese training paradigm and "freezing" one of the towers (meaning that no gradients are back-propped through it). The frozen tower computes keypoints which provide supervision for the non-frozen tower. The authors show good results on indoor and outdoor datasets. Some criticisms: The authors claim that end-to-end differentiability is important to simultaneously train the keypoints and descriptors and a core motivation of the proposed approach relies on the fact that selecting multiple extrema from a score map is not differentiable (line 33). But the empirical support for end-to-end differentiability isn't backed up by evidence. To back up this claim, the authors can add one simple ablation study where the descriptor loss is not back-propagated all the way through to the keypoints and compare the results. The evaluation uses too large of a pixel threshold (5 pixels) for computing matching score which makes it hard know if keypoints are precise. Precision of keypoints is important for many feature-based matching systems. Since the architecture is designed to output sub-pixel keypoint locations by using the softargmax (line 137), matching score should be evaluated at more strict thresholds which are less than 5 to provide evidence for this design decision. The indoor training component of the systems relies on depth maps for establishing correspondence. The authors acknowledge that depth measurements are often missing around 3D object boundaries (line 109), which is often true. Since image regions which do not have depth data are disregarded during training, this may create an unwanted bias away from detecting edges and corners, which are critical for traditional detectors such as Harris or ORB. The bias might then cause the detector to lack the ability to latch onto local textures. To address this, the authors can evaluate to the matching score at lower thresholds as described above. It would also be interesting to see the keypoints overlayed on top of the depth maps to see if the network is detecting any keypoints where the depth maps are undefined. The authors excluded an important result from Table 2: training indoors and testing outdoors. The authors cite the reason for doing so is the lack of photometric transformations in the indoor dataset (line 290). This may be true, but it doesn't justify excluding the result, as it may provide insight to further work which builds upon this. Selectively hiding results is bad science. In terms of clarity, the loss functions and network outputs are explained clearly, but the main Figure 1 could use some work. It's hard to tell that two images are being input to the diagram, possibly due to the lack of visual symmetry.

Reviewer 3



Overview: The paper presents an end-to-end deep neural network architecture that learns how to detect image features, estimate their scale and orientation, and compute a feature descriptor. Unlike existing architectures, the proposed network (dubbed LF-Net) uses 3D information (e.g., depth and relative pose) for the training phase. The paper presents experiments on publicly available datasets where LF-Net shows improvements over the hand-crafted features and competing architectures. Pros: - End-to-end architecture to learn image features and descriptors. - Formulation exploits 3D information (e.g., depth and relative pose). - Outperforms competing methods. Cons: - Clarity of the proposed architecture falls short. Detailed comments: Overall this is a good work and my main concern has to do with clarity. While I believe that the proposed architecture can detect interest points and describe them well, I found the text and Fig. 1 hard to follow. I think the paper is missing a detailed description guiding the reader about the flow of information in Fig. 1. This can help the reader reproduce this work and understand it better. I would suggest including D. DeTone, et al. SuperPoint: Self-Supervised Interest Point Detection and Description. CVPR 2018. to the experiments if possible, since LF-Net and SuperPoint address the same problem and are end-to-end.